

# Improving scheduling performance in congested networks

Arif Husen[1,2], Muhammad Hasanain Chaudary[1], Farooq Ahmad[1], Muhammad Imtiaz Alam[3], Abid Sohail[1] and Muhammad Asif[4]

[1] Department of Computer Science, COMSATS University Islamabad, Lahore Campus, Lahore, Punjab, Pakistan
[2] Department of Computer Science and Information Technology, Virtual University of Pakistan, Lahore, Punjab, Pakistan
[3] Department of Computer Engineering, University of Engineering and Technology Taxila, Taxila, Punjab, Pakistan
[4] Department of Computer Science, National Textile University, Faisalabad, Punjab, Pakistan

Corresponding author
Arif Husen, arif.husen@vu.edu.pk

## ABSTRACT

With continuously rising trends in applications of information and communication technologies in diverse sectors of life, the networks are challenged to meet the stringent performance requirements. Increasing the bandwidth is one of the most common solutions to ensure that suitable resources are available to meet performance objectives such as sustained high data rates, minimal delays, and restricted delay variations. Guaranteed throughput, minimal latency, and the lowest probability of loss of the packets can ensure the quality of services over the networks. However, the traffic volumes that networks need to handle are not fixed and it changes with time, origin, and other factors. The traffic distributions generally follow some peak intervals and most of the time traffic remains on moderate levels. The network capacity determined by peak interval demands often requires higher capacities in comparison to the capacities required during the moderate intervals. Such an approach increases the cost of the network infrastructure and results in underutilized networks in moderate intervals. Suitable methods that can increase the network utilization in peak and moderate intervals can help the operators to contain the cost of network intrastate. This article proposes a novel technique to improve the network utilization and quality of services over networks by exploiting the packet scheduling-based erlang distribution of different serving areas. The experimental results show that significant improvement can be achieved in congested networks during the peak intervals with the proposed approach both in terms of utilization and quality of service in comparison to the traditional approaches of packet scheduling in the networks. Extensive experiments have been conducted to study the effects of the erlang-based packet scheduling in terms of packet-loss, end-to-end latency, delay variance and network utilization.

## INTRODUCTION

Packet scheduling refers to allocation of sharing network resources among different types of flows with an objective to maximize the utilization and fair or ensure a targeted share for

each flow. It is a process of selecting packets for outgoing transmission based on certain criteria. Incoming packets are added to queues based on certain criteria depending on algorithm used and packets from these queues are selected for transmission on outgoing transmission link. Different packet scheduling schemes exhibit different delay, jitter, packet drop and throughput characteristics for users. Most of the time, the focus is to minimize the degradation of quality of service and maximize link utilization.

The existing approaches use different criteria such as priority, fairness, class of service and packet size for making the packet scheduling decisions. Although some of these parameters can be used to classify the packets efficiently, none of the existing parameters can represent the traffic characteristics in terms of capacity or volume. The capacity or volume of traffic characteristics across time and origin domains is generally used in the network dimensioning and capacity planning, but don't have any explicit control over controlling the scheduling of the packets in network nodes. The volume of traffic in a certain interval and origin is generally described in terms of Erlang units. Erlang represents the volume of traffic (TV) generated in a given busy period (BP). It was first described by Agner Krarup Erlang in 1908 (*Angus, 2001*). It has been widely used for network dimensioning and capacity planning in cellular networks and Private Automatic Branch Exchanges (PABX).

By calculating the Erlang traffic in an arbitrary period and origin, the traffic from different origins can be assigned in a way that higher Erlang values get more processing time as compared to low values. The allocation of the queue time in proportion to the traffic intensity provides efficient management of congestion and better utilization of network resources. The scheduling algorithm proposed in this paper utilizes above concepts to determine the period-based traffic intensity for each given origin of traffic. The proposed algorithm schedules traffic more efficiently according to Erlang capacities of traffic from different origins. The Erlang values when classified by origins are referred to traffic intensity (TI) of origin in a given interval. TI corresponding to a traffic profile of an origin is dynamically calculated by a capacity planning server. It can also be determined during the network development or planning phases using the network dimensioning processes. The distribution of Erlang profiles to different network nodes can be achieved in several ways such as a static configuration of network nodes and through dynamic communications using a TI server. In up-stream nodes of traffic flow, TI is accumulated to minimize the number of queues in core networks. The TI profiles can also categorize based on the type of service in order to realize the commonly known service-based scheduling so that Erlang values impact the scheduling decisions. The proposed algorithm can be used to implement integrated services or differentiated services mechanisms for controlling the quality of service. The focus of this paper is to present a novel approach to address the issues of network performance and utilization in congested conditions. Beside this, the following are major contributions:

- Provides a novel approach for scheduling decisions in networks with considerations of traffic planning and network conditions;
- Formulate a process to plan the traffic intensity profiles;

- Provide an algorithm to make and influence packet scheduling decisions;
- Present comparison results of performance and utilization;
- Identify potential machine learning based solutions for intelligent scheduling decisions.

## RELATED LITERATURE

The existing techniques of reciprocal processing of packets in queuing systems is based on priority, class, or a group of services for wired networks with objectives to either increase the fair access or some controlled access to the network resources. The wireless networks in this regard also focus on the conditions of the channels and energy conservation. Both wired and wireless networks employ common algorithms such as First-In-First-Out (FIFO), Round-Robin (RR), Deficit Round-Robin (DRR), Random-Early-Detection, General fairness (GF), Stochastic Fairness (SF) or service priority (SP) based scheduling algorithms. *Sungjoo et al. (2016)* have proposed a packet scheduling scheme that can employ multiple wireless networks on the outgoing side. In their proposal they have modified the TCP-friendly Rate Control (TFRC) algorithm, and they can distribute the packet transmissions to a group of users using multiple outgoing interfaces that increases the QoS by reducing packet loss in the process of scheduling or due to network congestion. Originally their proposal is in fact, an alternate approach to add the network resources or bandwidth to serve the same users, the result is optimization of quality of service, but it reduces the network utilization as more resources are added up. In contrast, our proposed work in this paper focuses on increasing the quality of services with given network resources and also optimizing network utilization. In fact, we can see that if we can enhance quality of service with given network resources, it will improve network utilization.

For wireless networks energy efficiency is an important research area. *Xu et al. (2016)* proposed a packet scheduling approach to maximize energy efficiency and it specifically focuses on wireless sensor networks where energy constraints are critically important due to the shorter battery lives of the sensor nodes. They have used an efficient offline packet scheduling approach along with a rolling-window-based online algorithm for scheduling decisions. Our proposed approach can significantly enhance the energy efficiency by unnecessarily comparative long periods of packets in queues and enhancing the quality of service. It can address the energy efficiency problems if we can intelligently correlate the traffic profiles of sending nodes, so that network nodes have necessary information to make scheduling decisions.

Binary search algorithm (BSA) is also an approach to enhance the packet scheduling. In this regard, *Pavithira & Prabakaran (2016)* has proposed a packet scheduling approach of Long-Term Evolution (LTE) based wireless networks using the novel binary search algorithm to increase the efficiency. It is noted that any such approach will lead to the degradation of the quality of service due to the fact that it incurs delays and jitter for the services in worst case and normal loads. In our proposed work we propose rather simple operations of constant complexity both in time and space. This cannot only

increase the network utilization and hence the efficiency, but also optimize the quality of service by minimizing the delay, jitter, packet loss, buffer overflows etc.

For video streaming applications, network coding selection is a possible way to improve the throughput and minimize the delay. Since it is difficult to synchronize users or peers, the bandwidth utilization degrades. *Huang, Izquierdo & Hao, 2016* have proposed an approach which coordinates between layer selection algorithms with a distributed packet scheduling algorithm to enhance the quality of video streaming. In our work, we shall show that both the improvement of quality of service and bandwidth efficiency is achievable without adding such complexities to the networks. Fairness is an important aspect that scheduling algorithms need to ensure among users, this is applicable to both with the best effort models and guaranteed service models. Users should be able to have access to a fair amount of network resources as per their subscriptions. *Deb et al. (2016)* has evaluated the relationship between the packet delay and fairness of network resource allocation among traffic flows or users. In our research we have shown that our proposed scheme can ensure the end-to-end delays with fair network resource allocation. In fact, in our proposed scheme, traffic profiles ensure the minimum number of network resources that shall be guaranteed across the network's flows or users or services.

*Deshmukh & Vaze (2016)* have proposed another approach for an energy efficient packet scheduling algorithm. They have used deadlines where a fixed number of packets is assumed to arrive. It assumes that all packets arrive within equal intervals. This assumption is basically difficult to arrive at in practice as the packet size and intervals may not be fixed rather different users send different packet sizes. In comparison, in our proposed approach rather than knowing the total number of packets within a fixed interval we use traffic intensities and intervals that can vary without hard-specified limits. Fair queuing schemes focus on to ensure reasonable fairness of allocation of network resources to the users or traffic flows, in practice the traffic characteristics changes, but users may send traffic up to their subscribed rates. *Patel & Dalal (2016)* have proposed a mechanism to adjust the weights assigned to traffic classes, that improves the performance in contrast to the fixed weights of Weighted Fair Queuing (WFQ) or Class Based Weighted Fair Queuing (CBWFQ). However, as the number of classes increases, the number of queues shall be increased so it may lead to scalability problems. In our proposed work, TIPS aggregate the queue on the upstream that minimizes the required number of queues. With a lesser number of queues, the performance of the network is increased, and complexity is decreased.

*Miao et al. (2015)* has proposed a preemption-based packet-scheduling technique to reduce packet loss and enhance the global fairness for software defined data (SDN). (*He, Xu & Luo, 2016*) evaluated the resource allocation problems for cases requiring preemption and queues are not properly processed. Due to this there is an increase in the length of queues and for preempted flows the delay is increased. In such conditions either it is required to decrease throughput for the preempted queue or add up delay for packets. Furthermore, the approach cannot handle cases where multiple packets fulfill preemption criteria. This condition is frequent in the upper layer nodes of networks.

Furthermore, this approach also does not consider time and origin-based problems of user traffic.

A dynamic core allocation to enhance the packet scheduling in multicore network processors is proposed by *Iqbal et al. (2016)*. They have considered a packet scheduling scheme that incorporates various dimensions of the locality to enhance throughput for network processors and minimize out of order packets. The specific problem addressed in this is the out-of-order packet arrival due to different processors handling traffic of the same flows. In our proposed work, the traffic originated from one source node is always handled in the same queue and this queue can be attached to a specific network processor and hence it can regulate the packet transmissions and minimize the out-of-order packet delivery.

*Sharifian, Schoenen & Yanikomeroglu (2016)* have proposed a scheme to handle the real-time and non-real time traffic in the common Radio Bearers (RB) for LTE based wireless networks. The improvements made by such an approach are augmented over all capacity. Such an approach is useful in access nodes; however the upper layer's networks have to rely on some other mechanism. In our proposed work we employ the role-based handling of the traffic, or we can use in-conjunction-mode where the access nodes can use a packet scheduling approach suitable to wireless medium or wireless technology to get relevant advantages.

*Mishra & Venkitasubramaniam (2016)* has presented a quantitative trade-off analysis between anonymity and fair network resource allocation. Accordingly, in anonymous networking, encrypted packets from various sources are re-ordered at routers randomly before processing in the outgoing direction. The authors have shown that it affects the fairness of network resource allocation. They have shown the results of First Come First Serve (FCFS) and fairness-based packet scheduling approaches using information-theoretic metric for anonymity and a common temporal fairness index that provides a degree of out-of-order packets transmitted. In our proposed work, the packets from the same sources are only allocated in the same queues within the specified busy periods and the change as Traffic Intensity (TI) indexes change hence through this way it ensures that packets are not always processed in the same queues, which indirectly provides anonymity.

*Yu, Znati & Yang (2015)* have proposed another approach for energy efficient packet scheduling through awareness of the delay experienced by the packets. We can see so far there are no proposals so far for the traffic intensity-based packet scheduling that employs the queue reordering not the packet reordering. *Han et al. (2015)* have proposed stochastic scheduling for optimal estimation of the parameters. In this regard, it is required to note that Stochastic Fair Queuing (SFQ) based packet scheduling approaches suffer from the issues of scalability, whereas in our proposed work we are proposing a scalable packet schedule scheme.

*Lee & Choi (2015)* have proposed a group based multi-level packet scheduling for 5G wireless networks. This approach considers the issue of user number and enhanced methods to reduce the interference between beams. This approach is in fact, an effort to address the issues of access nodes operating in the wireless domain. As with other

approaches of wireless enhancements-based algorithms TIPS can allow the in-conjunction mode operation where the wireless network related issues can be addressed and in upper layer nodes TIPS can provide better quality of service and network utilization.

There was another approach proposed for multi-resource environments that is a fair and efficient packet scheduling method called Active Time Fairness Queueing (ATFQ) (*Zhang et al., 2015*). The authors have recommended the approach for middleware devices in data center environments. In fact, this addresses the different processing time requirements on diverse resources. The issue of different processing time on middleware or server can be addressed with our approach by using a traffic profile suitable for related servers.

*Kaur & Singh (2015)* have proposed a Weighted Fair Queue (WFQ) based on Send Best Packet Next (SBePN) algorithm to enhance quality of service with a specific focus on multimedia application in mobile ad-hoc networks. With the WFQ based algorithms we have seen that in practice traffic intensities may vary, but with given weights either class based or service-based algorithms, the network may remain utilized and there may be higher traffic intensity traffic whose performance is subject to degradation. The TIPS algorithm assigns period-based weights according to the traffic intensities of source nodes and handles these weights by queue reordering and short consecutive resource allocation. We shall show in the results section that in fact, such an approach dramatically increases the throughput, reduces jitter and end-to-end delay, and optimizes the network utilization. This approach also reduces out-of-order packet arrival that is a typical problem at the receiver side where the received packets shall be required to re-order and application shall wait till all packets are arrived for a certain limit.

A class based weighted fair queue algorithm with generic traffic shaping mechanism is proposed by *Zakariyya & Rahman (2015)*. The class based weighted fair queuing is unable to respond to origin and time-based problems *i.e.*, time and origin of packets might be different but may carry the same classes, but class based weighted fair queuing will handle them in a similar way.

*Striegel & Manimaran (2002)* have proposed a scheduling technique that relies on signaling protocol for the resource reservation between end nodes; however, our proposed algorithm does not necessarily rely on any signaling for resource reservation; rather it works on scheduling packets based on planned or forecasted traffic intensities. In contrast to the signaling from end hosts for the resource reservation, the proposed algorithm uses the concept of Traffic Intensity signaling which is efficient in scalability and efficiency.

Differentiated services approach was a result of efforts to overcome limitations of integrated services and the basic idea in this was to aggregate traffic of similar characteristics based on bandwidth, delay guarantee requirements. In comparison, the proposed in this paper, although follows a similar methodology, but it limits the class or group-based decisions on access nodes and upper-level nodes relies on traffic intensity according to time, the region, and seasons.

Static Earliest Time First (SETF) and Dynamic Earliest Time First (DETF) approach, proposed by *Zhang, Duan & Hou (2001)* is an aggregation-based packet-scheduling technique that works based on the First-In-First-Out (FIFO) principle and uses

timestamps to schedule packets. These methods also do not consider traffic intensity, time and origin-based parameters in scheduling packets and nodes manage schedules in FIFO manner. Erlang based traffic Intensity has been extensively used for capacity planning on voice and Private Automatic Branch Exchange (PABX) systems (*Angus, 2001*) Public Switched Telephone Network (PSTN) Switches and Gateways. The commonly used Erlang distributions are Erlang B and Erlang C. For Erlang B, it is assumed that traffic packets are discarded when free resources are not available whereas the traffic packets are put in a waiting queue to be served upon the availability of the resources for Erlang C.

In this work, the Erlang type B and type C are transformed into packet transmission with arbitrary busy periods. Furthermore, the traffic intensities are classified based on their origin or serving area. Generally, the serving area is represented by an access node known as a Point of Presence (POP). The algorithm proposed in the article concentrates on a novel approach for packet-scheduling decisions aiming to improve the quality of service, network utilization for Congested Network Conditions (CNC) serving the different types of traffic generated by various sources with traffic intensity. This paper will show a comparison of performance, simplicity, scalability for Moderate Network Conditions (MNC). Erlang is widely used for network capacity planning and dimensioning of networks (*Angus, 2001*; *Choi, 2008*; *Dahmouni, Girard & Sansò, 2012*; *Davies, Hardt & Kelly, 2004*; *Glabowski & Stasiak, 2012*; *Guo, Zhang & Maple, 2003*) in telecommunication industry. This work highlights the significance of Erlang TI for minimizing the delay, jitter, packet loss and maximizing the data rates for CNC by using the TI dynamics in terms of time and origin-based characteristics for optimizing the scheduling. The proposed algorithm offers dynamic interval-based priorities of queues and proportionate scheduling decisions based on Erlang distribution. It allows making scheduling decisions based on the role of the nodes, origin, and time-based features of the traffic on each node. The roles of nodes are different and depend on network topology and architecture such as in a hierarchical network that uses different layers such as access, code, edge, and aggregation. The role of nodes determines how traffic is scheduled in upstream and downstream interfaces.

## MATERIALS & METHODS

The existing Erlang Traffic Intensity definitions are based on the assumptions for the Time Division Multiplexing (TDM) domain. In order to extend the concept for the statistically multiplexed traffic Internet Protocol (IP) networks, some terminologies are required to be defined specifically. This section discusses those definitions and description of various terminologies that are used to describe the working of the proposed algorithm.

### The role of a node

It is common to implement the IP networks in hierarchical layers where each layer provides dedicated functionalities. The nodes in these layers generally have different technologies and protocols that are used to perform certain functionalities. The hierarchical networks generally have the access network layer, aggregation/distribution network layer, core network layer, transport network layer and edge network layer.

In the telecom industry there are other layers such as the service network layer. The service layer deals only with the signaling traffic and hence has distinct characteristics and features. The service traffic also traverses the transport nodes in order to provide connectivity with physical distributed service nodes. The scheduling decisions are required to be made on each node; however, the objectives of scheduling vary per node.

The roles considered in this work include access, aggregation distribution node, core, transport, edge and service node. The names of different layers discussed may vary from operator to operator, however, some other names may have been used to represent the same functionality. The proposed algorithm uses the roles defined above to decide if the Erlang TI needs to be accumulated or distributed.

## Traffic

Since the IP packets may have different types of flows such as Transmission Control Protocol (TCP) or User Datagram Protocol (UDP) flows and some packets may not belong to any flow definition, however those packets play a role when scheduling them over the interfaces. So, for the purpose of this work, traffic is referred to as the packet stream having some common attributes like a source, destination IP/port addresses. The traffic definition considered here incorporates the effect of the packet size. In practice, there are different units that can be used to describe it but, in this article, the traditional *xBPS* routine is used. The x refers to K for Kilo, M for Mega, G for Gigabits per second.

## Traffic volume

In contrast to the traffic definition described in BPS, Traffic Volume (TV) refers to the aggregated count of bytes transmitted in an arbitrary interval and specific origin.

## Busy interval

Erlang TI is defined in terms of busy hours (BH). In this work in order to address the granularity issues the Busy Interval (BI) is any arbitrary interval over which the traffic intensity has been specified. The BI could be measured in seasons, week, day, hours are seconds and sub seconds. The values of TI depend on BI and TV and consequently the packet size in given traffic.

## Traffic intensity

With the extended definitions in the above sections, the traffic intensity represents the ratio between the expected volume of traffic and the actual capacity of the network or link in a given arbitrary busy interval and place in the network. Further details of the traffic intensities and methods to calculate them shall be covered in the following sections. For the end user point of view the traffic intensity is defined in the traffic volume generated by the user *vs.* the user subscription data rate in an arbitrary busy interval. In this article Erlang C is used to calculate traffic intensities of end devices. On a given time, there may be several services running such as IPTV, voice over IP and the high-speed internet. Considering σ the maximum data rate for normal operation of a service and active time be $h$, traffic intensity $t$ per service type is defined as below.

$$t = \sigma h$$

Furthermore with $q$ be the number of queues corresponding to a traffic type in given POP, $d$ the probability of delay for an arbitrary queue and packet generation according to Poisson process and delay as exponential distribution. The probability of delay of a packet is given by the following relation.

$$d = \frac{t^q}{q!}\frac{q}{q-t} \cdot \frac{1}{\left(\sum_{i=0}^{q-1}\frac{t^i}{i!}\right) + \frac{t^q}{q!}\frac{q}{q-t}}$$

## Point of presence

The term Point of Presence (POP) represents, in fact, the area of an access node of the network where subscribers are connected directly or indirectly. POPs can be classified based on origin, network function or layer such as Residential POP and Commercial POP represents the service node that serves the users in a residential area and commercial areas respectively. Similarly, access POP, aggregation POP and core POP represent the functionalities provided by nodes in the access layer, aggregation layer and core layer respectively. Other categories follow a similar method.

## Traffic profile

Traffic profile is a representation of the traffic intensity of a user or node in the form of a histogram that has the BI, Packet Size, and other related parameters. These profiles are used by the proposed algorithm to determine the information of the arbitrary intervals and intensities and packet sizes. The TPs are aggregated on the upstream nodes while distributed on the downstream nodes. There are different methods to determine the TP that includes the Forecast Based (FOB), Historical Network Usage Based (HNUB), Application Signaling Based (ASB) and Operator Policy Based (OPB). The details of these methods shall be covered in the next sections. However, for the purpose of evaluation and comparison in this work, the FOB method has been used to determine the traffic profiles. FOB provides the upper bound to traffic estimates (represents what networks are designed to support) as compared to the ASB or HNUB that represents a more real state of the network.

## Time and origin characteristics

Different users in different types of serving areas have different network usage patterns. Some POPs could be seasonal, others could be generating traffic in different hours of the day or week based on geographical statistics. Furthermore, there would be different characteristics of the traffic generated by the different POPs in terms of the packet size, latency requirements and data rates. All these characteristics are referred to as TAOC characteristics and mainly used in traffic profiling processes.

## Scheduling constants

Scheduling Constant SK and K depends on the traffic profile information of POPs. These constants are used to control the scheduling decisions and may also represent policies. The SK constant is linear additive to administratively manipulate scheduling behavior and is configurable per node whereas the constant K is defined globally on a network or layer domain.

## Congested network conditions and moderate network conditions

The congested network conditions (CNC) represent a network condition where maximum users generate the traffic over a network. It is the situation where POP has high activity. Generally, not all POPs always operate in CNC state. There are only some periods that may have this condition. This fact allows the network provider to define their oversubscription policies based on the activity of a POP. On other hand, moderate network conditions (MNC) refers to a POP condition that has moderate or less than maximum activity and users in those POPs may generate traffic that is intermittent or sparsely distributed over the timescale. In this state, free resources available on the network exceed the traffic load. The networks remain under-utilized because operators generally plan the network according to the CNC requirements. The proposed algorithm considers these considerations and helps to control extravagant investments on the network infrastructure to reduce the cost of network services.

## Comparison parameters

The evaluation and comparison of the effects of using the proposed algorithm will be based on the certain parameters that includes QoS measures such as sustained throughput delay, latency and jitter and network utilization. These parameters shall be measured with the proposed algorithm and with other existing algorithms over a single network topology that consist of the hybrid simulated and emulated environments. The details of topology and environment shall be discussed in the subsequent sections.

## THE PROPOSED ALGORITHM

This section covers the details and operation of the proposed algorithm called Traffic Intensity based Packet Scheduling (TIPS). It uses the TI profiles for scheduling decisions with consideration of various factors such as the node role, scheduling factors and TAOC. There are few parameters that need to be defined for description of the algorithm. These includes the $Q$, $sQ$, $BI$, $TI$, $SBI$, $SQ$, $enQ$, $deQ$, $QL$, $SC$ and $xQ$ that represents a packet scheduling queue, special queue for control purpose, buys interval, the traffic intensity, selection procedure for busy interval, queue selection procedure, procedure to add a packet to queue, removal procedure from a queue, queue length, scheduling criteria and the queue selected by the $SQ$ procedure respectively. The TIPS algorithm exploits a set of traffic profiles on nodes. On the down-stream it distributes the profiles into multiple profiles based on the traffic information and on upstream nodes it aggregates the profiles received from the other nodes. Through this way the number of queues is reduced consciously as traffic traverses the upper layers of the network hierarchy and increases the

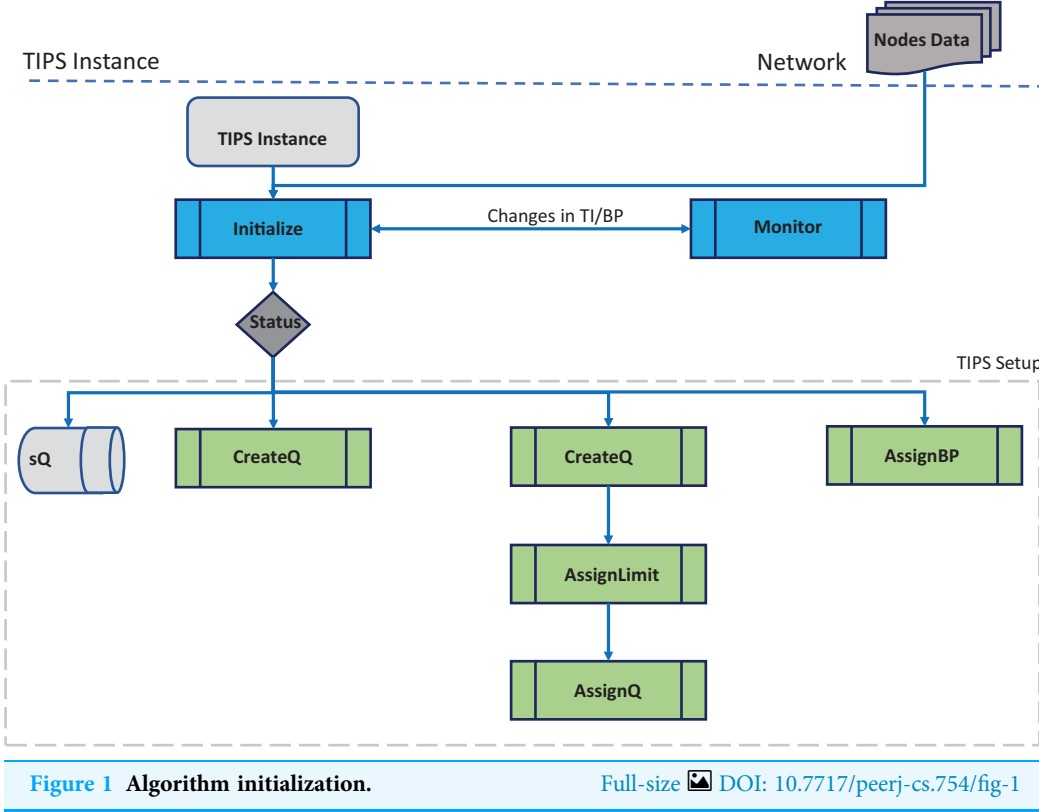

**Figure 1 Algorithm initialization.**

number of queues on the downstream nodes or nodes facing the users. The lesser number of queues in upper layer nodes reduces the processing delays and time spent whereas on the downstream nodes it allows the queues as required by the user types. The traffic profiles on the network nodes can be configured in several ways as enlisted below.

1. Add traffic profile database to each network node statically by administrator
2. Deliver the selected set of traffic profiles from a profiling server
3. Allow upstream nodes to learn or receive profiles from downstream nodes

Figure 1 shows the initial setup phase corresponding to the Initialize ( ) procedure TIPS algorithm pseudo code (Algorithm I). In this phase, a number of queues are created according to the traffic profiles received or configured on the network node. The queue limit is determined based on the traffic intensity values in the given busy interval and these limits change with changes in the BI. Different queues are created with
a size determined by the TI values for each profile on each node running the TIPS algorithm. On the upper layer nodes of the network hierarchy, profiles are either received from the downstream nodes or fetched from a local node database or any other method as discussed in the earlier section. This step is important as it defines the number queues and their size. In addition to this, the user traffic queues, a special queue is created to handle the packets generated for the control protocols that carry various information such as routing updates, link or channel setup, resource reservation messages. In the upstream direction the profiles are aggregated and on downstream nodes, the

| Algorithm I | | |
|---|---|---|
| Initialize ( ) | RECEIVE_SIDE(Packet P) { | TRANSMIT_SIDE( ) { |
| { | **VAR** ln ← LastNode( ); | **VAR** p ← Packet *P; |
| Bind_local_variables ( ); | **VAR** bp ← get_BP( ); | **VAR** p ← Determine Profile; |
| Load_profiles ( ); | **VAR** p ← get_Profile( ); | **VAR** bp ← Determine BP; |
| Initialize_queues ( ); | **VAR** ps ← get_PS( ); | **VAR** nr ← Determine NodeRole; |
| VAR Special queue. | **VAR** nr ← get_NodeRole( ); | **VAR** tg ← TrafficGroups; |
| **VAR** GLOBAL TI; | **VAR** tg ← get_TrafficGroups( ) | **VAR** ql ← GET_SORTED_QUEUE-LIST (DESCENDING TI); |
| } | **VAR** qto ← ENQ_TO; | **VAR** q ← SELECT Serving_Queue( ); |
| | **VAR** sCriteria ← Evaluate SCriteria( ); | **VAR** i ← GET_SCHEDUALAR_Instance; |
| | **IF** (sCriteria == True) { | |
| | Qs(P); | **IF** (**SIZE**(Qx) < TI) { |
| | } | DDQ(Qx); |
| | ELSE { | } |
| | **IF** ( SIZE(qto) > TI) { | ELSE { |
| | Drop Packet( ); | Qx->DEQUEUE( ); |
| | } ELSE { | } |
| | qto ->ENQUEUE(P); | SET_NEXT_QUEUE; |
| | } | RETURN *p; |
| | } | } |

profiles are distributed according to the number of profiles. The special queue is processed immediately if its size is greater than zero. Furthermore, any other queue that is being processed is preempted and the *enQ* and *deQ* operations are performed on the special queue. The special queue can be considered as the highest priority queue with preemption. All the elements that are contained in the TIPS setup block are used for the initialization of the TIPS algorithm. During the initialization of TIPS, nodes have *sQ*, current BI and different queues according to the traffic profiles.

In each BI, the enQ and deQ processes functions according to the information in the profiles available. The size of the queue is chosen according to the volume of the control traffic existing in the network and it is ensured that control packets do not encounter any packet loss. Receive side processing of the TIPS algorithm is shown in Fig. 2. On the reception of a user packet the TIPS check if there are any control packets, if the *sQ* is empty then the received packet is processed. For the user packets received on the interface, packets are processed by *enQ* according to the *QL* limits. If the *QL* exceeds the packets are immediately dropped without any further considerations processing, however, such a situation rarely occurs as while the packets are being enqueued during the BI, the deQ processes are also working to empty the queue according to TI values. This process is repeated on each incoming interface of the node. The processing of the TIPS algorithm on the outgoing side is shown in Fig. 3 where *sQ* is de-queued with high priority. If the size of

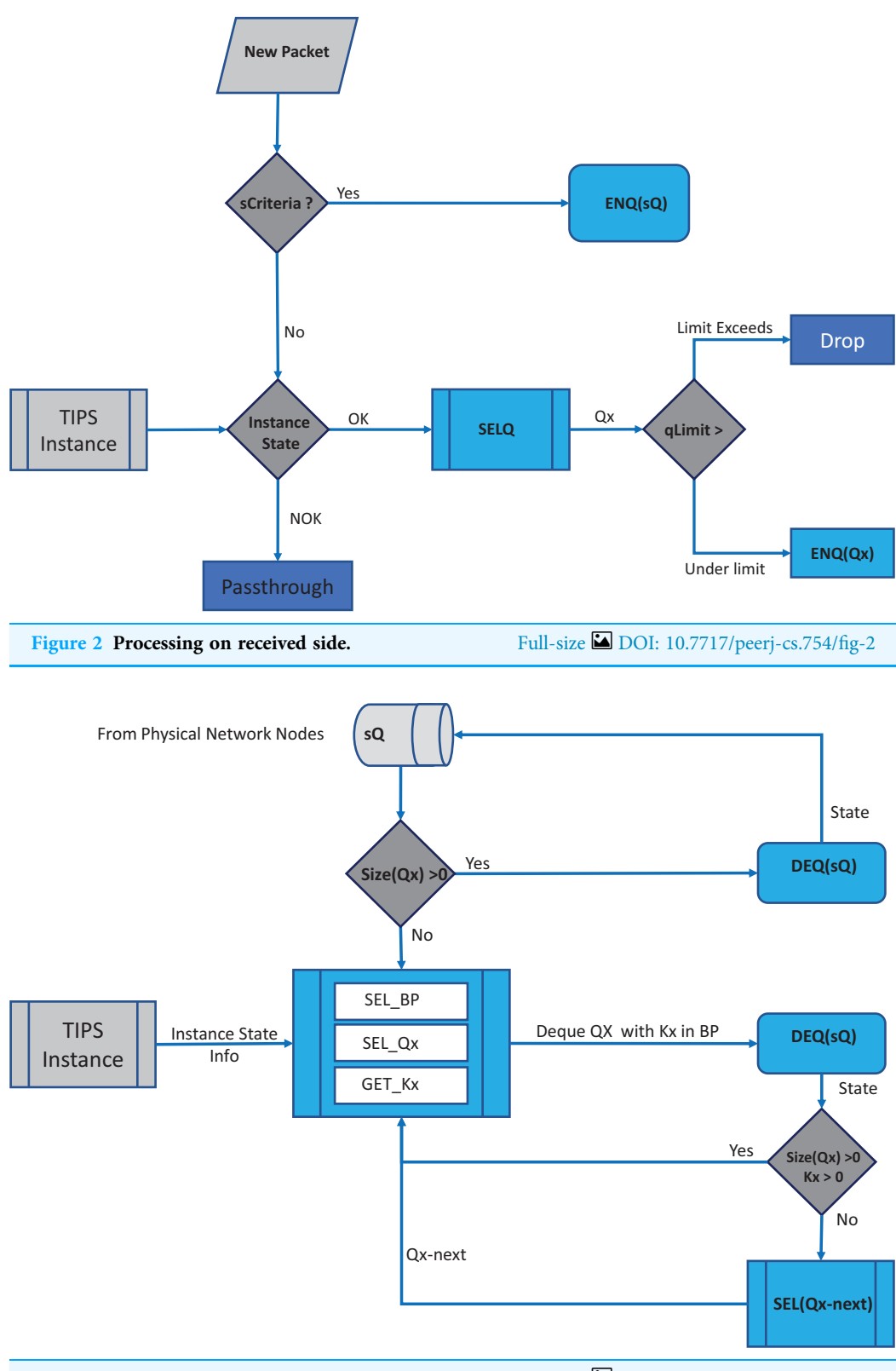

**Figure 2  Processing on received side.**               

**Figure 3  Processing on transmit side.**               

*sQ* is zero, the queues having higher sizes are processed first according to the number specified by the algorithm. The current size of any queue is determined by the TI values, so higher the TI values for a given queue, a greater number of packets shall be processed repeatedly to equilibrate the queue system. During the *deQ* processes the *SK* and *K* parameters play an important role that defines the limit of the packets that can be processed by deQ regardless of the TI values. Generally, these factors are the upper limit to the TI values hence eliminating the chances of the non-processing of other queues.

If a *Q* is *deQ* by a limit specified by the TI values or the upper limit of SK, the next Q is selected by the SQL for deQ processing, and this process is repeated until all queue size reaches zero.

During packet departure, $SK_x$ (where $SK_x = SK\,(ij)$, $i = Q$ number and $j = BI$) confirms the traffic with higher TIs does not interfere with the latency of queues having the smaller values of TI. It also makes sure that the traffic corresponding to the high values of TI is allocated sufficient resources.

## METHODS OF TRAFFIC PROFILING

In the earlier sections, different traffic profiling techniques have been mentioned. This section focuses on specific details of each method and discusses how these can be used for the proposed algorithm.

### A. Forecast or business planning based

This is the simplest method of building traffic profiles. This is similar to the procedures as used in the capacity planning of the network where the objective is to identify the services, the number of users, future margins, and over-subscriptions, Service Level Agreement (SLA) for each of POP in the given network. Generally, these are the forecasts part of the business plans of an organization. The typical classifying is based on the marketing product being offered and the relevant SLAs. The parameters that required converting a forecast to profile are as followed:

- Number of subscribers per service per POP
- Bandwidth per service
- Service classification

Based on the demographic survey, it provides the following four values for each service for each POP.

- Hour wise Usage (HWU)
- Day wise Usage (DWU)
- Week wise Usage (WWU)
- Month wise Usage (MWU)

Apparently, this looks to be a very tough task of suggesting the usage for each hour of the day of the week of the year. For HWU, the important thing that we need to know is to type POP. If POP is residential and the community is a city, and man and woman are

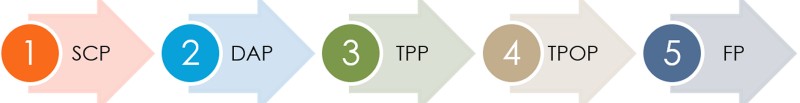

**Figure 4 Process steps for HNUB traffic profiling.**

employed, then we simply need to know the official working hours. In working hours residential usage will be low, whereas the commercial POP will have higher usage. For DWU, the major task here is identifying the working days and weekend days, holidays. In case of WWU, we differentiate the weeks based on any specialty attached to it, if nothing important is scheduled then simply choose it as 100%. For MWU , the month level specialty differences required, if nothing different, then simply fill it with 100%.

## B. Historical network usage based

This technique is based on monitoring a given network for its network utilization. Monitoring is required at the access nodes, and the upper layer nodes shall use the fundamental concepts of profile aggregation. The traffic monitoring can be achieved with open-source tools like CACTI (*Ian et al., 2012*) and MRTG (*MRTG, 2012*). The things that need to be monitored included the following:

- Required information is access interface utilization, packet size accords the time scale.
- The minimum, maximum and average packet size.
- Identify the traffic classes or groups if required.
- Obtain the policies for traffic groups

The process of building the traffic profiles five pages as shown in Fig. 4. It consists of five steps as discussed below.

### Statistics collection phase

This phase requires the collection of hour wise interface utilization, subscriber information, services information, network topology & service prioritization and bandwidth guarantees.

### Statistics analysis phase

This phase involves the categorization of access POPs, defining the usage patterns with respect to time (hour, day, week, month, season, day, and night), defining the usage patterns with respect to POP type and analyzing the packet sizes.

### Traffic profiling phase

This phase calculates the traffic volumes per hour per day per week per month/day/night/season/special event, determine the average packet size, calculate the Traffic intensities, define the distinct busy periods, and define the average packet sizes per BI.

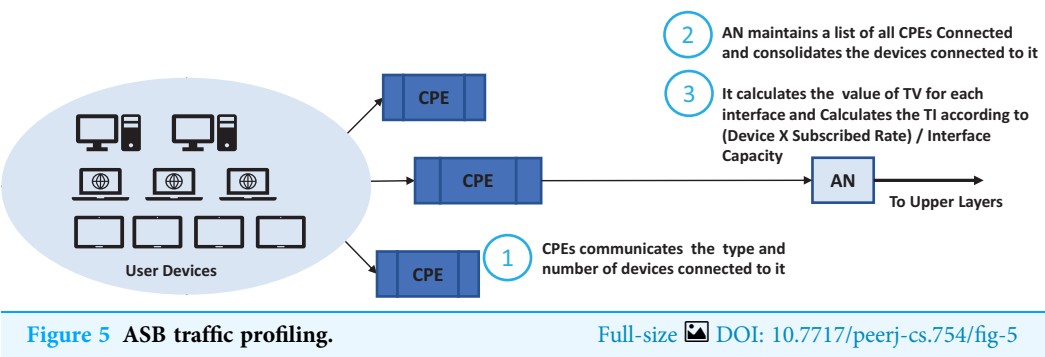

Figure 5  ASB traffic profiling.     

## Profile optimization phase

In this phase the traffic policies like priority and bandwidth guarantees per service, time specific policies to traffic profiles, origin specific policies to traffic profiles, administrative policies to traffic profiles are applied to the output of the previous phase.

## Final phase

Using above information this phase is used to generate traffic profile files and apply to network nodes as part of the TIPS.

## Application signaling based

In this case, we use an application based signaling protocol to inform the current status of the client machine by analyzing the following.

- Current running applications
- Over-the-Air (OTA) settings
- Operating System (OS) update settings
- System up down behavior

We look at the networks enabled, running applications and their capabilities such as if it can initiate a voice session, file transfer, browsing, video session and file download. It will monitor when a session started, file download started, conferencing started, email retrieval started, retrieval frequency. In this scheme an agent is installed on each host that communicates with access nodes updating its current expected usage. An agent monitors the user's behavior, OTA settings, browsing behavior. We used a scenario where the user hosts communicate with the Customer Premises Equipment (CPE) which is generally a xDSL Modem or Wi-Fi Router. Router maintains a list of running applications and host capabilities which is updated by the agent. Generally, in residential cases the user's hosts may have mobile, notebooks and desktop, gaming, or IPTV. The simple case may be just to enumerate the number of devices and their capabilities and apply common basic characteristics to each of them. We take this case in this study where the CPE just reports the user devices and their capabilities. The ASP traffic profiling scheme is shown in Fig. 5 and its integration with the nodes running TIPS is shown in Fig. 6. CPE may collect the traffic details such as applications and their expectation and network usage pattern and communicate with the TIPS instance running the access

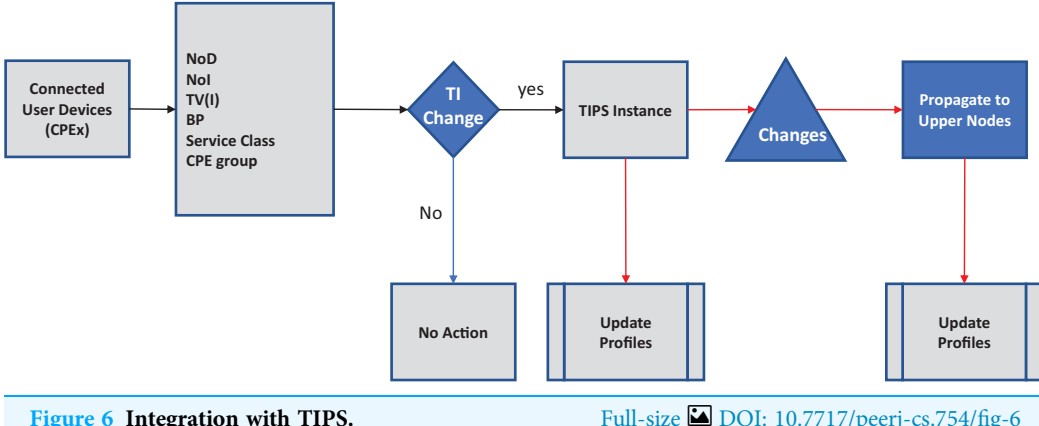

**Figure 6 Integration with TIPS.**     

node and provide information such as total interfaces, traffic volume, traffic intensity, service classes and a group per interface. This information is compared with existing traffic profiles associated with the CPE and if the changes are found the profiles are updated and communicated to other network nodes also. The subsequent scheduling decisions shall follow the updated profiles to reflect changes in the traffic characteristics.

# IMPROVING THE SCHEDULING PERFORMANCE

The TIPS algorithm operates on the fact that POPs generally exhibit different characteristics in terms of usage and users have different applications that generate different types of traffic. In addition to the above, different POPs have different characteristics in terms of the time when traffic is generated by users and the type of region. Furthermore, the network operators are interested in having control over the behavior of unexpected traffic originating from users. The existing commonly used packet scheduling techniques such as Deficit Round Robin (DRR), Round Robin (RR), Priority Queueing (PRQ), Fair Queue (FQ) and Class Based Fair Queuing (CBFQ) do not exploit these considerations in their operations. Consequently, the network providers opt to use additional techniques to control the scheduling in network nodes. Similar is the case with scheduling schemes for wireless networks that considers channel conditions and other factors. Due to the absence of above features in packet scheduling techniques following consequences are observed.

- The techniques are not aware of capacity and network dimensioning constraints
- The commonly used algorithm cannot cope with the diversity in traffic characteristics
- Their objectives are relative remain static
- Existing techniques cannot cope with the dynamic nature of the traffic.

The absence of these features degrades the resource allocation fairness and quality of service parameters such as End-to-End Delay (E2ED) and its variance incurred by unpredicted traffic management in queues with static criteria, for example, the priority queuing is unable to distinguish between traffic generated by different POPs having common priority. Furthermore, it also cannot distinguish traffic with the same priority on

different timescales. Similar is the case with round-robin or deficit round-robin and weighted fair queuing or class based weighted fair queuing as all traffic classification, their types and mechanisms of fairness are fixed over time and for each POP. In order to address these limitations, the proposed TIPS algorithm is devised to consider all these aspects and offer improved scheduling for a different quality of service parameter such as packet loss, delay, jitter, and sustained throughput.

## A. Management of unpredicted delays

In $BI_x$, the TIPS algorithm chooses the highest TIs available in traffic profile, and enQ corresponding packets in $Q_x$. After this, it selects next highest TI and enQ its packets in $Q_{\{x+1\}}$. This process repeats till all traffic profiles are processed and all packets are enQ to corresponding queues. The arrived packets are dropped if the corresponding size of the queue exceeds TI value and the SK factor. At any given time, packets in all queues determine the total size of the queues ($S_Q$) and is equal to the sum of all TIs from all POPs.

$$S_{q,bp} = \sum_{i=1}^{n} TI_i \tag{1}$$

If $S_{q,bp}$ surpasses QL then packets belonging to the corresponding profile are dropped where TI is exceeded. For the deQ case, the maximum limit of the count of packets that are allowed to remain in a queue is the product of the average packet time and sum of all TIs. In the experiments, it is seen that the maximum time a packet occupies in a queue is five to seven times of their packet-size. The minimum time occupied in any queue is most of the time equal to one packet time.

## B. Regularizing the traffic bursts

The spikes in network traffic generally increase in delay while making decisions of packet scheduling. In cases the decisions are priority based, or class or round-robin based then such a situation could lead to packet loss. In the proposed algorithm the spikes in traffic are regulated with allocation of the length of the queue in proportions to size of TI. The Traffic having the same class or priority or group with different origin or time get distributed allocations.

## C. Handling the large packets in traffic

Larger size packets lead to an increase in E2ED due to their large transmission time and occupancy time. The proposed algorithm inherently solves this issue with a cautious profiling of TI in each BI. TI values are based on traffic volume that considers the aspects of the size of the packet's users, sessions, or duration. Due to this the TIPS algorithm is capable of effectively controlling the additional delays due to large size of packets and jitter normalization.

## D. Variations in delay

Generally, end-to-end delay variations occur in the traffic due to different network conditions and non-linear distribution of decision of scheduling. The proposed algorithm

TIPS handles this problem by successively dequeuing packets according to TI values. Since the TI values are directly dependent on the, hence the proportionate processing time is allocated to different types of the packet leading it to decrease delay variations. Furthermore, the proper planning of traffic profiles helps to eliminate such variations.

### E. Minimizing the packet loss

Packet loss in the network is due to various reasons. One of the reasons could be inefficient scheduling decisions where the queues are filled rapidly, and limits are reached. In such cases the network nodes have no option but to drop further packets. The dropping of few packets part of connection-oriented flows leads to impact on the congestion control mechanism adopted by the upper layers of the TCP stack. Furthermore, the large spikes of packets with lesser priority for PQ and lesser weight traffic for CBFQ don't get enough resources and queues quickly overflow that lead to the loss of packets. Due to the burst regularization in the TIPS algorithms, larger packets get resources according to their calculated TI values. This leads to reducing the probability queue overflow that subsequently minimizes the packet loss due to scheduling issues.

### F. Role based scheduling

In hierarchical topology of networks, nodes are arranged in layered fashion, such as access, distribution, or aggregation and code nodes, where each layer has specific network functionalities. The proposed algorithm provides the node role-based scheduling mechanism. The scheduling decisions in access nodes differ from other types of nodes. On access nodes, TIPS can be used to provide the class-based or group-based scheduling. These classes are groups that can be handled differently according to the needs of the class of groups. These groups or classes must be matched to the traffic profiles in order to benefit from this feature. On the nodes that implement the aggregation role, the TIPS algorithm makes decisions by classifying the traffic based on TI values received from other nodes. This reduces the queues count and simplifies the processing of queuing operations. With core roles it aggregated and redefined TI values. These aggregated values are then used in the making decisions for the scheduling of packets. This directly results in lowering the queues further and minimizing the processing delays.

### G. Shorter queue sizes

The proposed algorithm regularizes the traffic bursts by manipulating TI values. The short burst of the packets causes momentary rise in TI values and consequently more processing time allocated in such cases. This leads to minimal queue growth and maintains the average queue size within the capacity of interface buffers. Generally, shorter queue sizes or the lower rate of queue growth improves the throughput, minimizes the delay, and jitter.

### H. Joint mode support

The proposed algorithms support the joint operation with other scheduling techniques. The joint operation is based on the different nodes *i.e.*, some nodes in the network may be using the traditional scheduling algorithms and other nodes use the TIPS Scheduling.

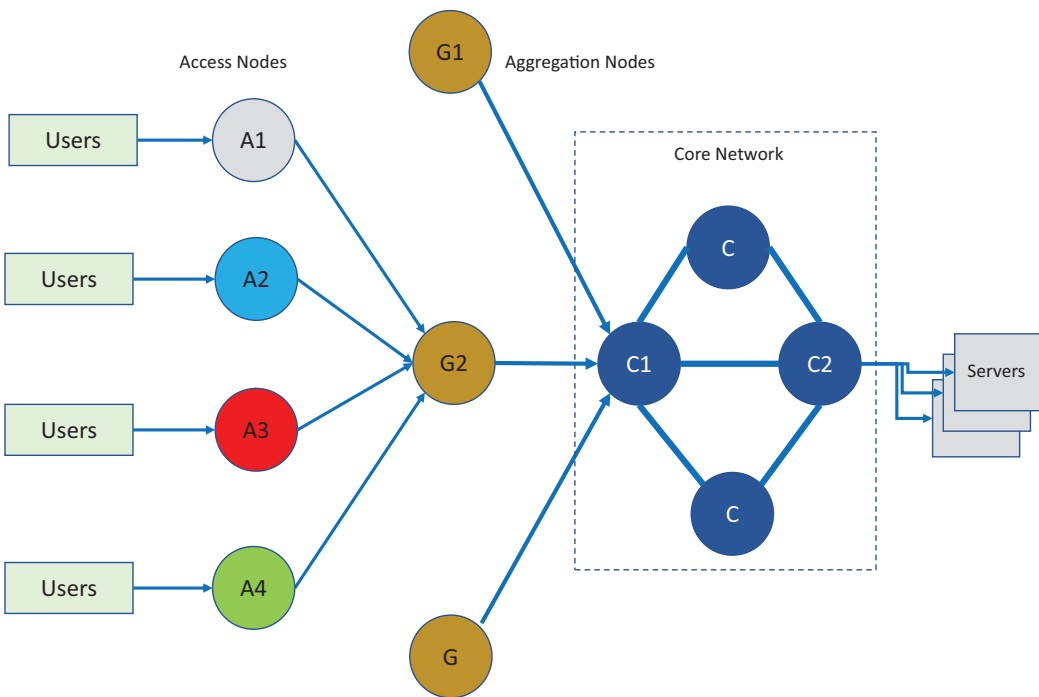

**Figure 7 Network topology for evaluation and comparison.**

With this support the objectives like prioritized handling or class-based handling of traffic can be retained.

## EXPERIMENTAL SETUP FOR EVALUATION

For evaluation of the proposed scheduling technique, A setup consisting of network simulation using Network Simulator (*NS2, 2012*) and containers (*Dockers, 2012*) was used. The end points were created in the Linux environment where each user and server were created standard docker containers. The traffic generated by the container hosts was connected to the access nodes in the NS2 simulated environment. The host containers generate traffic according to a selected scenario type of POP. The traffic generated by the hosts follow a random pattern following the BIs and given traffic intensities. The real-time traffic passes through core, aggregation and access nodes with topology as shown in Fig. 7 and the respective bandwidth parameters are given Table 1. The network topology was created in a simulated environment. The simulated network nodes run the TIPS instances with statically configured roles as per topology of the network that follows the three-layer hierarchy commonly used in practical networks. In the analysis of the results of the TIPS algorithm, the color scheme as shown in the figure will be used and traffic traversing over the links will be defined with node colors. For example, the traffic generated on node $A_1$ will follow black color, $A_2$ will be blue and similarly for the remaining nodes. For IEEE 802.3 based scenarios as shown in topology, following interface capacities are defined both for CNC and MNC models.

**Table 1 Link capacities.**

| # | Links | Max bandwidth (Kbps) |
|---|---|---|
| (a) - For CNC scenario | | |
| 1 | A1 . G2 | 25,000 |
| 2 | A2 . G2 | 25,000 |
| 3 | A3 . G2 | 25,000 |
| 4 | A4 . G2 | 25,000 |
| 5 | G1—ÀCN1 | 25,000 |
| 6 | G3—ÀCN1 | 25,000 |
| 7 | G2—ÀCN1 | 25,000 |
| 8 | CN1—ÀCN2 | 50,000 |
| (b) - For MNC scenario | | |
| 1 | A1 . G2 | 25,000 |
| 2 | A2 . G2 | 25,000 |
| 3 | A3 . G2 | 25,000 |
| 4 | A4 . G2 | 25,000 |
| 5 | G1—ÀCN1 | 50,000 |
| 6 | G3—ÀCN1 | 25,000 |
| 7 | G2—ÀCN1 | 25,000 |
| 8 | CN1—ÀCN2 | 75,000 |

**Table 2A Distribution of traffic volumes.**

| BI | $N_1$ | $N_2$ | $N_3$ | $N_4$ | $N_5$ | $N_6$ |
|---|---|---|---|---|---|---|
| $BI_1$ | 2,985 | 404 | 1,992 | 746 | 2,522 | 3,511 |
| $BI_2$ | 1,950 | 1,840 | 1,538 | 1,008 | 3,733 | 2,124 |
| $BI_3$ | 1,406 | 877 | 1,504 | 1,971 | 1,885 | 3,480 |
| $BI_4$ | 1,543 | 2,418 | 817 | 942 | 3,006 | 2,101 |
| $BI_5$ | 171 | 1,674 | 1,082 | 2,081 | 756 | 3,349 |
| Sum | 8,055 | 7,213 | 6,933 | 6,748 | 11,902 | 14,565.00 |

## Application for the traffic generation

A custom-built model application (MAP) is used to produce traffic as per BIs, TIs and s for different nodes. MAP is intended to operate in 24 BIs and s. Each BI has a distinct TI allocated for each node that is calculated using the FOB approach discussed in the earlier sections. The theoretical volumes of traffic in packets shown in Table 2A for each of the notes for the experimental setup. The packets are set to 1 kB, which may vary in practical applications. In such cases the total bytes transferred are divided by the average for a common base for comparison. Above table shows traffic volumes for five periods; however, the simulation is run for several periods. Table 2B represents the start busy interval that ends with the beginning of the next busy interval. Although the tables show the same values for five busy intervals having different traffic volumes, the experimental setup is run for several busy intervals. The equivalent TI derived for these intervals is shown in Table 3. The process of derivation of TI values is discussed in previous sections.

**Table 2B FOB BI distribution.**

| BI | $N_1$ | $N_2$ | $N_3$ | $N_4$ | $N_5$ | $N_6$ |
|---|---|---|---|---|---|---|
| 1 | 0 | 0 | 0 | 0 | 0 | 0 |
| 2 | 2 | 2 | 2 | 2 | 2 | 2 |
| 3 | 4 | 4 | 4 | 4 | 4 | 4 |
| 4 | 6 | 6 | 6 | 6 | 6 | 6 |
| 5 | 8 | 8 | 8 | 8 | 8 | 8 |

**Table 3 Equivalent TI distribution.**

| BI | $N_1$ | $N_2$ | $N_3$ | $N_4$ | $N_5$ | $N_6$ |
|---|---|---|---|---|---|---|
| $BI_1$ | 12 | 5 | 8 | 5 | 10 | 14 |
| $BI_2$ | 8 | 9 | 6 | 4 | 15 | 8 |
| $BI_3$ | 7 | 3 | 7 | 8 | 7 | 14 |
| $BI_4$ | 6 | 10 | 4 | 4 | 12 | 8 |
| $BI_5$ | 7 | 4 | 9 | 3 | 14 | 5 |

The main objective in this experimental setup is the delivery of the volume of traffic efficiently and reliably without loss of packets and optimized quality of service and performance parameters such as delay, jitter, and highest possible throughput. Table 3 shows Traffic intensities corresponding to each BI. It shows TIs for the first five BIs and for the remaining BI, the TI are calculated in a similar way. The TI table follows the relationship as per Eq. (2) where $TI_{ij}$ is the intensity of traffic for node j in busy interval i, $TV_{ij}$ is the volume of traffic in node j and the busy interval I and SK values are constant. $TV_T$ is the traffic volume of outgoing links on nodes.

$$TI_{ij} = \begin{cases} 1 + SK & 0 > (TV_{ij}/TV_T) < 1 \\ (TV_{ij}/TV_T) + SK & (TV_{ij}/TV_T) > 1 \end{cases} \tag{2}$$

The scheduling fact SK values chosen are 2,1,2,1,2 and 2 respectively for N1 to N6. The values of this parameter as derived using the Eq. (3) where $TI_{ij}$ refers to the traffic intensity allocated to $i_{th}$ queue in $j_{th}$ BI, and $SK_{ij}$ represents the SK values for $i_{th}$ queue in $j_{th}$ BI. The $i_{th}$ queue is assigned dynamically to $i_{th}$ profile in the corresponding busy interval.

$$SK_{ij} = \begin{cases} 0 & (TI_{ij})/\left(\min_{i=1,\ldots,q_n}(TI_{ij})\right) < 1 \\ (TI_{ij})/\left(\min_{i=1,\ldots,q_n}(TI_{ij})\right) & otherwise \end{cases} \tag{3}$$

In the experimental setup the K constant is set to 1 for all the nodes of the network.

## RESULTS

As discussed earlier, the experimental setup uses the docker container for creating the communication end points and NS2 for the network functionalities. The MAP application

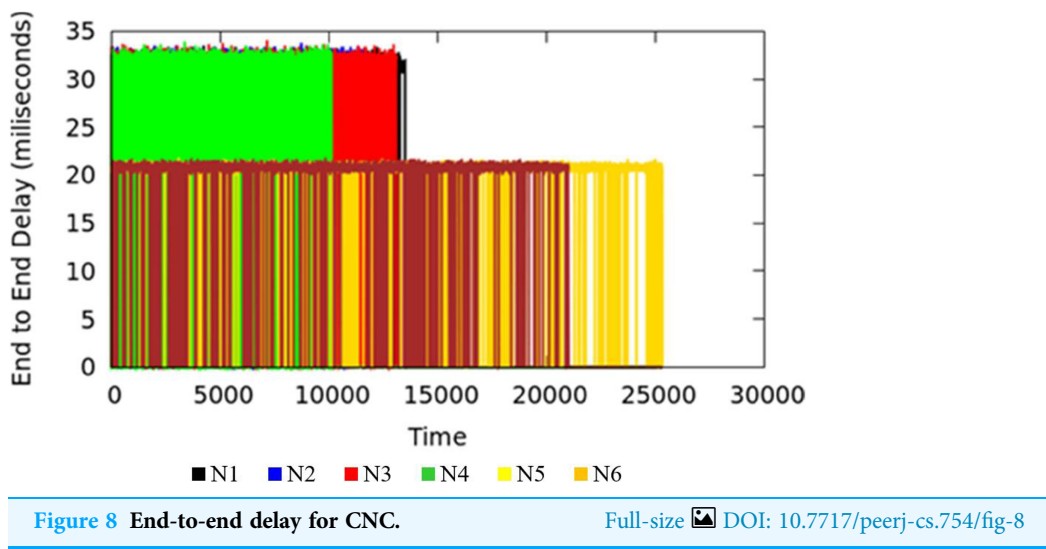

**Figure 8  End-to-end delay for CNC.**     

is used to generate traffic corresponding to the traffic profiles. The type of traffic was set to real-time RTP.

## A. Performance in congested network conditions

This section discusses the measurements carried out for TIPS performance in congested network conditions. The network setup was created with the link allocation as given in the earlier sections. On CNC, TIPS algorithm performance is significantly better as compared to traditional scheduling techniques.

*Measurement results of delay*

End-to-end Delay is an important parameter in the measure of quality of service of the applications. The end-to-end delay in the network could be due to various reasons where queuing delay is one factor. The TIPS algorithm is intended to minimize the delay incurred due to the waiting time of packets in queues. Waiting time in the queue for packets can be lowered by considering the rate of packet origination from source nodes and adopting the scheduling behavior accordingly. Large values of traffic intensity require a faster processing of corresponding packets in comparison to smaller values of TI. The applications running on the host machines may have similar end-to-end requirements but different TI values in different BIs may increase the delay in the scheduling process for lesser and higher TIs. The TIPS algorithm manages the queue allocation time by selecting suitable profiles in the queues and manipulating the order of enQ and deQ functions. Figure 8 shows the measurement results of end-to-end delay exhibited by the TIPS algorithm in the CNC case. It depicts the latency experienced by packets in the network in milliseconds (ms) including the delay due to the propagation in the medium. Some nodes like Black, Red, Green, and blue are at a distance of three hops and other nodes with brown and yellow color are at a distance of two hops. In the experimental setup the propagation delay is set to the value of 10 ms for all links.

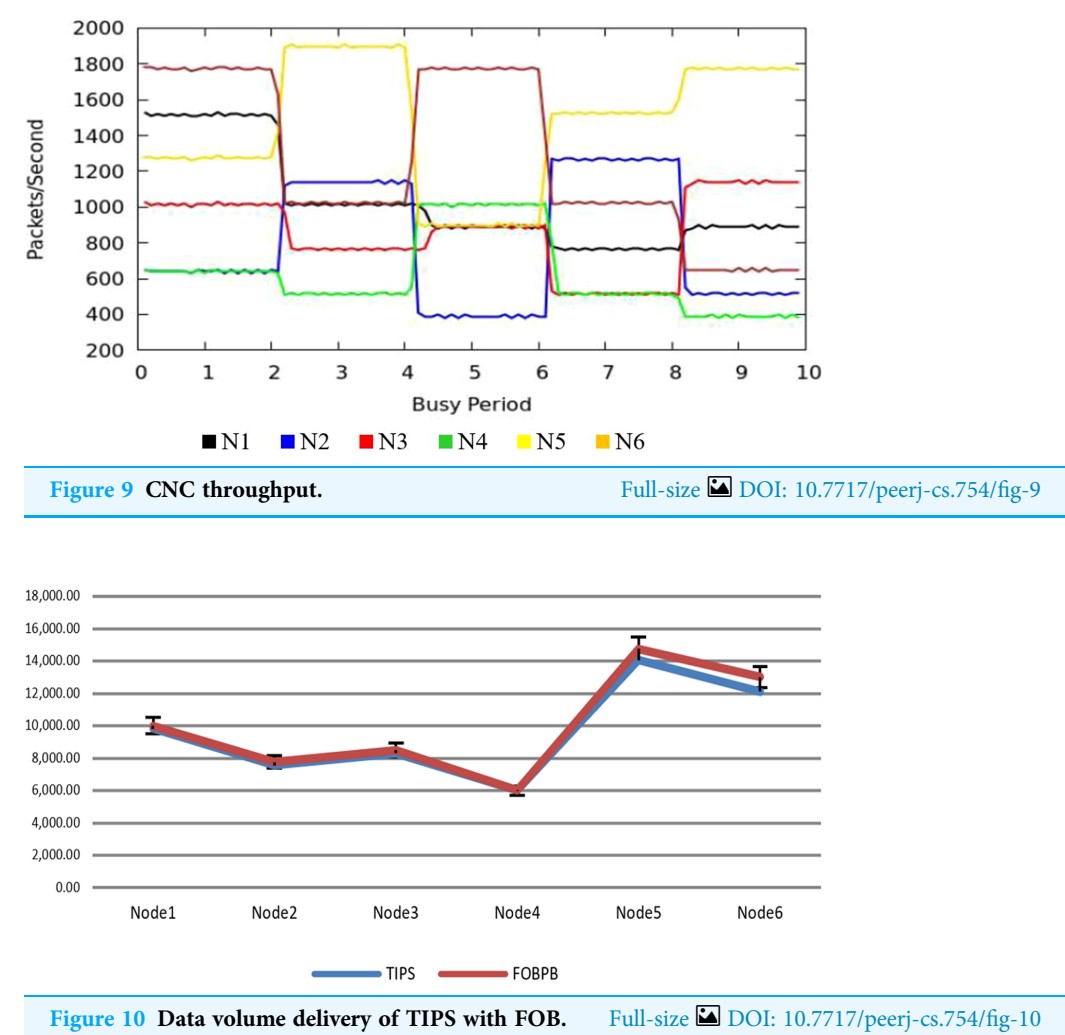

**Figure 9** CNC throughput.

**Figure 10** Data volume delivery of TIPS with FOB.

### Results of throughput measurement

The Fig. 9 shows the measurement results of sustained throughput attained with the TIPS algorithm. It represents the packets- per-second (PPS) on y-axis and timescale on *x*-axis. The significance of these results is the smooth curve of throughput in every busy interval. The traffic was generated using a Model Source Application (MAP) that generates the Real-Time Protocol (RTP) traffic following the deterministic TI distribution. The comparison of the FOB traffic volumes and the actual traffic volumes successfully delivered is shown in Fig. 10. FOB traffic volumes are calculated from hypothetical forecasts of services and their expected usage pattern. It represents the sum of packets on the *y*-axis and the respective source node on the *x*-axis. The overhead of the MAP application is about 3.6% that includes the packets sent/received for the keep-alive and acknowledgements during the lifetime of the session. The figure shows that the TV received is exactly 3.6% lower than the FOB traffic volumes and the TV Delivery ratio is 96.43% as shown in the figure.

### Delay variance or jitter comparison

The delay variance (DV) is also a significant factor that can degrade the QoS of service for applications running on communication endpoints. It is a significant parameter for the jitter buffer and synchronization of multimedia endpoints. As discussed in *ITU-T-G.8261.1/Y.1361.1 (2012)* and *RFC-3339 (2012)*, the jitter is defined as the variance in end-to-end delay experienced by a given node between two successive packets. For the experimental evaluation, the Eq. (4) is used to derive the jitter in the received traffic where $D_i$ and $D_j$ is delayed between two successive packets i and j.

$$J_{ij} = Abs(D_j - D_i) \tag{4}$$

For DV or jitter, the average, minimum and maximum values are derived using the Eqs. (5)–(8). For these equations the reordering of the packets has been ignored.

$$J = \frac{abs(D_j - D_i)}{j - 1} \qquad i \neq j \tag{5}$$

$$J_{avg} = \sum_{n=1}^{pc} \frac{abs(D_j - D_i)}{(j-1)*pc} \qquad i \neq j \tag{6}$$

$$\lfloor J \rfloor = \frac{abs(D_j - D_i)}{(j-1)} \qquad i \neq j \tag{7}$$

$$\lfloor J \rfloor = \frac{abs(D_j - D_i)}{(j-1)} \qquad i \neq j \tag{8}$$

In the above set of equations, pc refers to the count of packets, Abs refers to the absolute value delay experienced and i and j are the sequence number of packets. Figure S1 shows the jitter or delay variance in the traffic received by different nodes. The values of jitter of each packet in the experiment duration were recorded and the values shown in the figure are instantaneous. This helps to clarify 1st-order and higher-order jitter for networks. The Figs. S1A–S1F represent the jitter in the traffic originating at N1 to N6 respectably. The values are shown in ms units and it shows both the negative and positive jitter values. The negative jitter values represent the packet arriving in lesser duration in comparison to the last packet *e.g.*, the change in the variance is negative. The figure shows that smooth jitter is experienced by the packets originated at the node 1.

The N1 is located at a distance three hops from the destination node. The results show that there is steady DV in every BI of the traffic profile. Some BIs have significantly low jitter in the range of 0.1 ms to 0.3 ms. The other nodes show steady DV up to 0.5 ms. As with N1 results it shows both the positive and negative values of jitter. The negative results show the delay was lesser than when the last packet arrived. N2 is located at a distance of 4 hops from the sink node. This is observed that N-2 also shows the same pattern as N1, however, each BI has dissimilar values as compared to N-1. For N3, both of the positive and negative values of variance are shown where the negative values represent the packet with shorter delay as compared to the last packet received. The N4 is located at a distance of 3-hops from the sink node. It has maximum variance up to 2 ms.

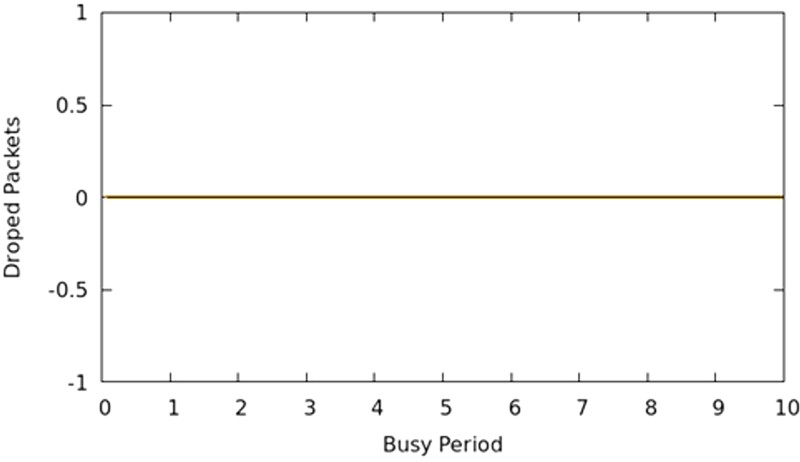

**Figure 11 CNC packet drop for all nodes.**

The maximum value for N4 is only for few packets and average jitter, a pattern like other nodes. The N5 is located at a distance of 2 hops from the sink node. It can be observed that it shows quite steady jitter for N5 during the experimental transmission. The N6 is located at a distance of 2 hops from the sink node and it follows a similar pattern as N5. In this section, a term steady jitter has been used. The steady jitter is useful in many ways. It eliminates the presence of t of higher order variances such as 2nd order jitter.

### Packet drop characteristics

Packet loss is likewise a significant parameter that depicts the performance of networks. Greater the packet lost; the more are the re-transmissions on the network. Various methods exist to effectively react to re-transmission requirements as opposed to retransmission of the entire portion of information such as selective-repeat and TCP-Reno. This article focuses on limiting the packet loss because of buffer overflow or queue overflow in the network with proficiently scheduling decisions. The packers are not allowed to wait in queues to their max limits irrespective the class, weight, or priority excluding the case where network allocations limits are reached, moreover, it guarantees an appropriate command over which packet should be dropped if the system is working in CNC state. In this experimental evaluation, it used a MAP application that produces traffic in such a manner that creates the CNC conditions of the network, and traditional scheduling techniques begin to drop bundles. They drop packets as well as queuing time overshoots to high values. Using the TIPS algorithm, the CNC conditions are handled with a suitable intensity aware method to guarantee that packets are not dropped unless traffic limits are reached or resources exhausted on the interface. Figure 11 shows that packet drop results for all nodes. It shows that there was not any packet loss at any point in the experimental network in CNC. Other traditional approaches of the scheduling cause an irregular packet loss for given traffic profiles because of the surpassing of the queue limits. Nonetheless, since the TIPS processes the queues according to TI values, queue limits remain within limits.

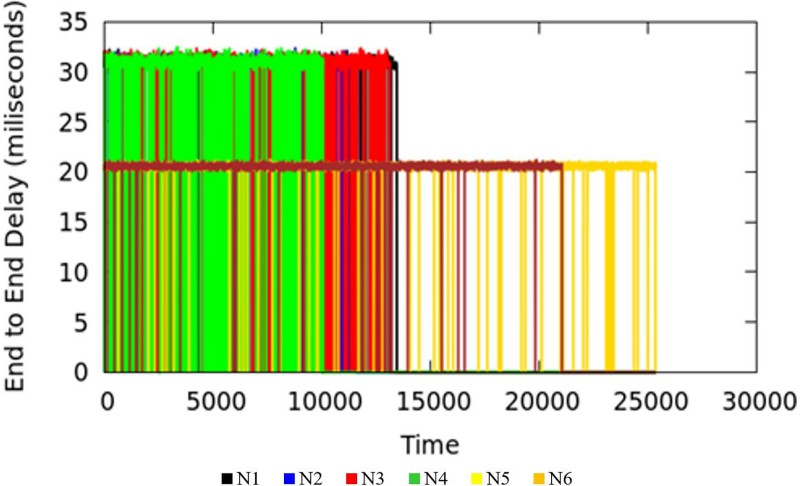

**Figure 12 End to end delay in MNC.**

## B. TIPS in moderate operating network conditions
### *Measurement results of E2ED*

E2ED is the difference in time of packet creation at a source node and the reception time at a sink node. One side delay is called a one-way delay in contrast to the RTT that shows time of upstream and downstream sides. E2ED also includes time consumed in packet transmission at source nodes, its propagation, other processing at nodes and the waiting time in queues. The Eq. (9) shows the calculation method for the end-to-end delay where $D_i$ represents the delay and $T_{i+}^S$ represents the time when a packet is created and $T_{ir}^R$ represents the time when it is received at destination.

$$D_i = T_{i+}^S + T_{ir}^R \tag{9}$$

Equation (9) $i$ is $i$th packet sequence, $+$, $r$ represents the packet addition to the queue at the sending side and the packet reception at the receiver side respectively. In the experimental setup, the propagation delay is set to a constant value of 10 ms. In the comparison section, it will be shown that in MNC, E2ED generally remains near to propagation delay value. However, if more traffic arrives, and the network gets into the CNC state, the E2ED due to queuing processes becomes significant and values increase beyond the propagation delay of single hop. Figure 12 represents the E2ED of traffic originating in access nodes in MNC conditions. The different colors in the figure represent traffic originated in the source nodes.

### *Throughput results*

Figure 13 illustrates the throughput of traffic originating in access nodes in MNC. It represents the PPS (Packet Per Second) on *y*-axis and timescale on *x*-axis. The colors refer to the sustained throughput of different nodes in different BIs. The Comparison of Figs. 9 and 13 shows that the proposed TIPS algorithms exhibited similar results, both in CNC and MNC. The throughput in MNC shows that required data rates are achieved. Other scheduling algorithms that are compared with TIPS algorithms also offer similar

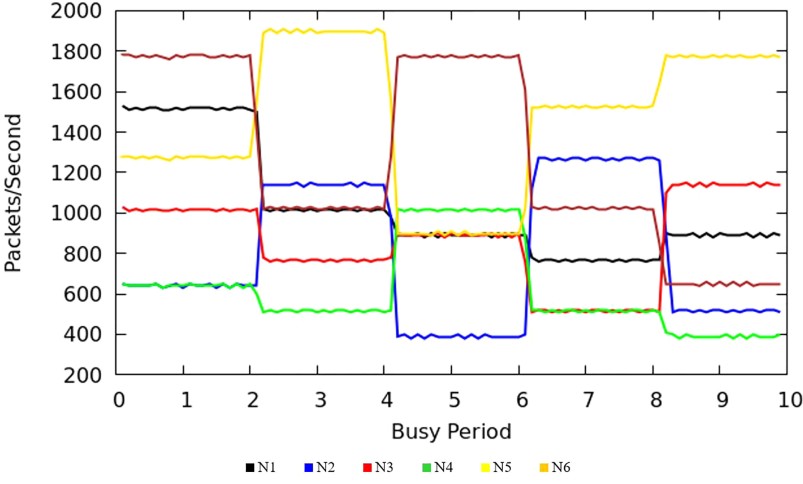

**Figure 13 Throughput in MNC.**

results in MNC. But with CNC, data rates were significantly degraded with all other scheduling approaches.

### End-to-end jitter

As with the case of CNS, the jitter in MNC also follows the finest limits and average values remain less than 1ms with spikes of 2 ms. Figures S2A–S2F illustrates the jitter in ms units experienced by traffic originated at N1 to N6 respectively. It shows both negative and positive values where the negative values show that packet delay is less than the last sequence of the packet. It can be observed that it has a single hit of jitter surpassing 1.5 ms while normal jitter remains under 1 ms limit. It can be observed that N2 has two spikes of jitter more than 1.5 ms and the average value remains less than 1 ms limit. The N3 experiences multiple spikes of jitter with more than 1.5 ms limit and the average jitter remains at 1 ms. N4 and has a single spike of jitter with maximum value more around 1.5 ms and the average jitter remained below 1 ms. For N5 and N6 the jitter remains below 0.6 ms. These two nodes are two hops away, whereas the first four nodes were at a distance of 3 hops from the destination. In a general network state, it can be observed that delay variance remains around 0.3 ms for each hop.

### Packet drop behavior

Similar to the CNC results there was no packet loss observed for all nodes. However, it will be shown in next sections that in CNC, The TIPS algorithm was able to achieve higher throughput and hence has higher traffic volume delivery.

## DISCUSSION

### Comparison with other algorithms

In order to compare the TIPS algorithm performance with others, the DRR, RR, FQ, RED, and SFQ were also tested on identical network and traffic profiles as used for the TIPS algorithm evaluation.

| Table 4 Packet loss results and comparison with TIPS. | | | | | | |
|---|---|---|---|---|---|---|
| | % Packet drop (Total) | | | | | |
| Algorithm | $N_1$ | $N_2$ | $N_3$ | $N_4$ | $N_5$ | $N_6$ |
| DRR | 18.6 | 7.86 | 4.44 | 3.79 | 2.25 | 1.4 |
| FQ | 49.93 | 24.42 | 5.07 | 14.08 | 6.46 | 12.66 |
| RED | 11.39 | 8.22 | 7.31 | 9.97 | 4.39 | 6.25 |
| RR | 10.57 | 8.29 | 7.94 | 15.27 | 2.57 | 3.73 |
| SFQ | 26.3 | 7.05 | 2.07 | 3.38 | 6.47 | 12.44 |
| TIPS | 0 | 0 | 0 | 0 | 0 | 0 |

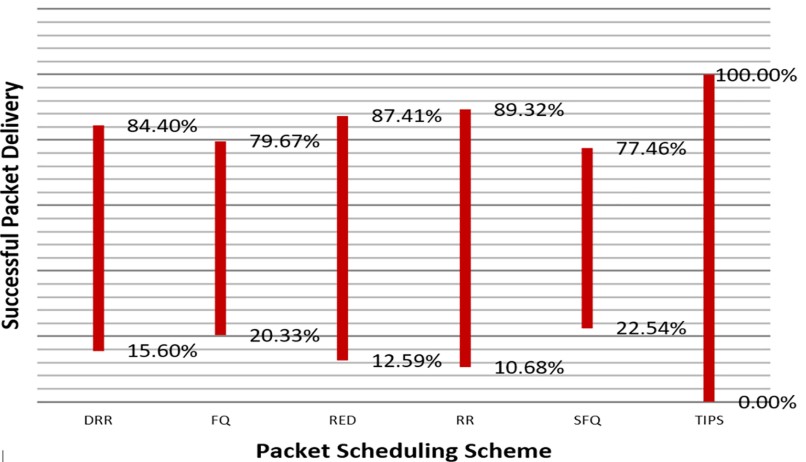

**Figure 14 Data rate in MNC.**

## A. Packet drop behavior

Different existing scheduling algorithms such as DRR, RR, FQ, RED, and SFQ were simulated on the experimental network. The measurement results were captured both for the CNC and MNC. It can be observed from the results that the TIPS algorithm provides a significantly improved performance in comparison to the traditional schemes of scheduling. In the CNC state, there is an obvious distinction in results of these approaches when contrasted with TIPS.

Table 4 gives a synopsis of the rate of packet drop in CNC. The execution of all different methodologies debased essentially though TIPS created the same outcomes as in the event of MNC. Packet loss greater than 1% on a network significantly degrades the performance of real-time services and proves to be fatal for critical and interactive services that require the responses to be received within stringent limits. For other applications, greater packet losses decrease network utilization and efficiency due to frequent re-transmissions. The frequent retransmission leads to further aggravation of congestion in the network.

Figure 14 illustrates the comparison of the successful packet delivery rate to the destination. The red line refers to the successful delivery of packets. It shows a comparison

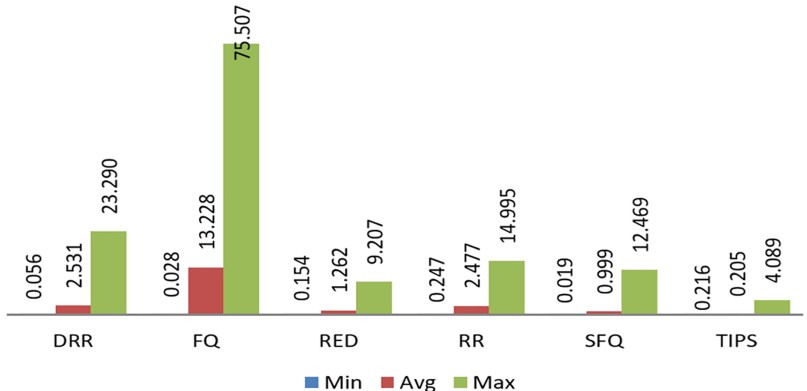

**Figure 15 Comparison of delay in milliseconds experience by the traffic.**

of successful delivery of generated packets at source and packet drop with different various packet scheduling algorithms. The SFQ has the lowest successful delivery ratio of packets due to higher packet losses. On other hand, the proposed TIPS algorithm exhibits the maximum successful delivery of generated packets. These results show that the TIPS algorithm provides high efficiency in CNC and increases the network utilization, minimizing congestion and providing optimization of overall performance.

### B. Comparison of E2ED in CNC

This section provides the comparison of E2ED improvements with TIPS algorithm with other traditional scheduling techniques. The method to calculate the E2ED has been also discussed in previous sections. The E2ED comparison is from three different aspects such as the average, minimum and maximum delay experienced by the packets from the source to destination over different nodes in CNC conditions. The comparison of these aspects is given in Fig. 15. It shows the comparison of E2ED maximum, minimum, and average values in ms units as experienced by the traffic originated in different nodes. The delay values in these figures are not inclusive of the propagation delay. N1 to N4 are at a distance of three hops from destination and N5 and N6 both are at a distance of two hops. It can be observed that the traffic experiences least delay in CNC with the TIPS algorithm where the maximum value of the delay experienced is limited to 4 ms and average values are in fractions of ms. The worst case can be observed for FQ where E2ED is beyond 50 ms and the average delay remains around 20 ms. This comparison shows that the performance of all these schemes degrades in CNC state in comparison to the TIPS algorithm where performance is observed efficiently for the E2ED.

### C. Comparison of throughput in CNC

The comparison of packet throughput attained with all the algorithms is presented here. The throughput is calculated as the fraction of packets sent by the source and packet received by the respective sink nodes. The throughput relationship follows a relationship as shown in the Eq. (10) where $P_t$ represents the packet throughput, $R_i$ represents the

| Table 5 Maxmimum throughput. | | | | | | | |
|---|---|---|---|---|---|---|---|
| **Algorithm** | **Percentage throughput** | | | | | | |
| | $N_1$ | $N_2$ | $N_3$ | $N_4$ | $N_5$ | $N_6$ | **Sum** |
| DRR | 86.84 | 96.05 | 98.42 | 99.6 | 99.78 | 99.9 | 97.3 |
| FQ | 65.14 | 86.3 | 97.17 | 92.16 | 99.77 | 99.9 | 88.4 |
| RED | 95.12 | 95.34 | 95.86 | 93.28 | 99.78 | 99.9 | 97.4 |
| RR | 95.82 | 95.12 | 95.49 | 90.51 | 99.77 | 99.91 | 97.1 |
| SFQ | 84.72 | 95.92 | 98.18 | 97.54 | 99.78 | 99.91 | 96.7 |
| TIPS | 100 | 100 | 100 | 100 | 100 | 100 | 100 |

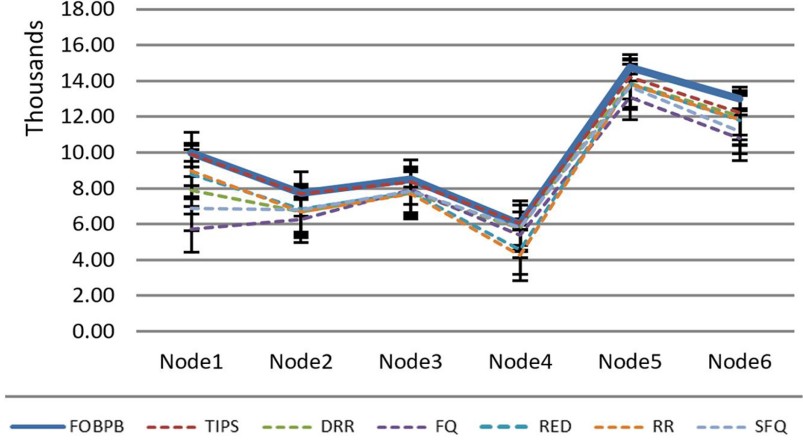

**Figure 16 Traffic volume comparison with TIPS.** 

packets received by node $j$ sent from node $i$ and $S_{ij}$ represents the packets sent by node $i$ to node $j$.

$$P_t = \left( \frac{\sum\limits_{i=1}^{n} R_{ij}}{\sum\limits_{i=1}^{n} S_{ij}} \right) * 100 \tag{10}$$

The Table 5 shows that the TIPS algorithm is generally better to give 100% data delivery. This fact depends on packets sent by the source nodes and packets received by the sink node. Since there was no packet loss as we saw in previous sections, the TIPS algorithm gave significantly better throughput while delivering the data in CNC conditions. These outcomes have a direct relationship to packet loss in the network. The worst instance of packet loss was with the FQ approach that has the lowest throughput of effective delivery rate. Figure 16 illustrates the analysis of the data delivery with different algorithms in comparison to the FOBPB and Fig. 17 shows throughput with different algorithms normalized to TIPS. Furthermore, the comparison of the actual delivered amount of data is shown in Fig. 18. It is evident from these results that the TIPS algorithm provides the best data delivery rate in comparison to all other scheduling techniques and

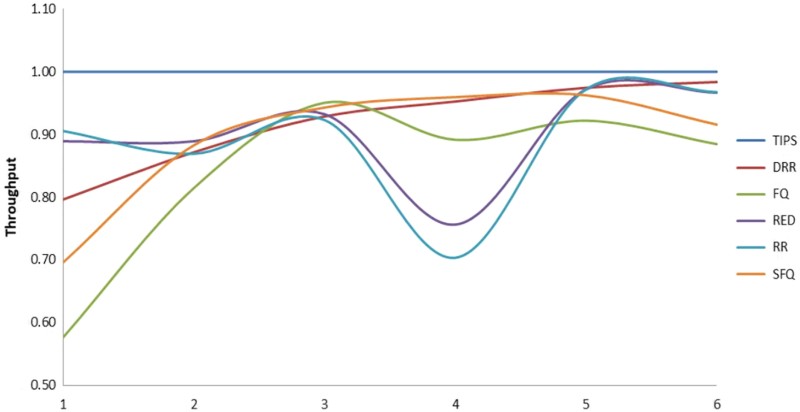

**Figure 17 Comparison of throughput normalized to TIPS.**

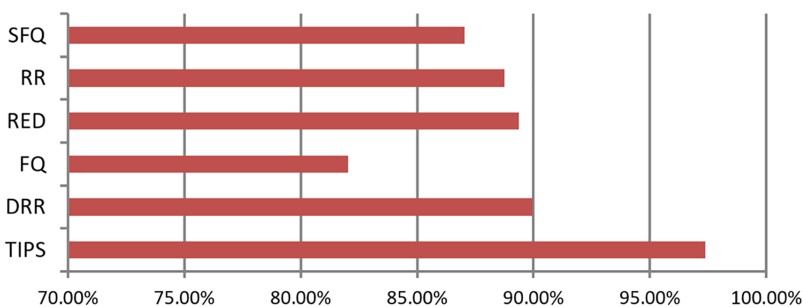

**Figure 18 Delivery percentage of traffic volumes.**

**Table 6 Jitter comparison.**

**Packet Jitter (milliseconds)**

| Maximum | Method | Average | Method | Minimum | Method |
|---------|--------|---------|--------|---------|--------|
| 2.09 | TIPS | 0.35 | TIPS | 0.11 | TIPS |
| 3.73 | DRR | 0.4 | DRR | 0.11 | DRR |
| 2.74 | FQ | 0.42 | FQ | 0.11 | FQ |
| 3.01 | RED | 0.36 | RED | 0.16 | RED |
| 3.59 | RR | 0.42 | RR | 0.32 | RR |
| 3.77 | SFQ | 0.4 | SFQ | 0.31 | SFQ |

the max target data volume determined by the FOB process. It provided 96.5% successful data delivery ratio and if adding 3.6% MAP overhead it achieved approx. 100% data delivery ratio.

### D. Comparison of delay variance

The comparison of delay variance seen by packets from different nodes during the experiment is shown in Table 6. The maximum variance faced by packets with the TIPS algorithm is 2.09 ms with a maximum distance of 3 hops. The average value of variance is

0.35 ms and minimum variance is observed around the value of 0.11 ms. With the TIPS algorithm the maximum jitter is minimized by 30–45% in comparison to other scheduling algorithms. It shows significant improvement to the data delivery achieved with the TIPS algorithm.

## OTHER ALGORITHMS

There are some other scheduling schemes such as class-based queuing (CBQ), Class based Weighted Fair Queuing (CBWFQ) and Priority Queuing (PQ) that provide some additional approaches to assign the weights or priority to the different packets based on some user criteria. These algorithms work fine for prioritizing traffic classes or groups, but these suffer from issues if traffic characteristics are changed. Additionally, the limited number of classification methods puts limits on their performance. Furthermore, these algorithms support only static configurations and cannot cope with the changes in the traffic characteristics dynamically.

## APPLICATIONS

We have seen in previous sections; the TIPS algorithm takes advantage of diversity in traffic characteristics across nodes. The typical application scenario is the outgoing interfaces on aggregation and core nodes of networks. The underlying reason is that one such application involves a number of POPs. With the traditional approaches the rule configuration may become complex. Another case is to utilize it in aggregation and core nodes or regional and national core nodes in telecommunication IP/Optical Transport Networks. Similar logic applies here as above as there is diversity in traffic originating from different. In web edge hubs, TIPS can essentially make distinction in deferral, jitter and parcel drop difficulties. Other potential applications incorporate WAN connections in big business organizations, where TIPS can give better execution and organization usage because of lower deferrals, dormancy, and bundle drops. Higher these qualities the lower network execution and organization usage as end clients would retransmit the lost or dropped packets or breaks because of enormous measures of delay.

## FUTURE RESEARCH DIRECTIONS

With the advancements in artificial intelligence and service requirements, the networking domain is also required to use the machine learning based techniques to help the autonomous and cheaper and faster decision-making in managing the network functionalities. There are various ML based approaches that can predict the traffic forecast based on the analysis of the previous historical usage of the data. These algorithms can extract the Spatio-temporal insights from the data and make predictions for the future. Furthermore, there are ML based techniques that can provide the efficient traffic classification considering the various aspects of previous network usage. With the machine learning based approaches the TIPS shall be able to work efficiently. Extreme learning, the Bayesian network and decision tree can classify the internet traffic with the focus on peer-to-peer applications (*Sena & Belzarena, 2012*). The traffic classification algorithms must provide classification information to other systems as soon as possible to make

necessary decisions. Support vector machine based Early Traffic classification models can address this issue (*He, Xu & Luo, 2016*). The ML models like Multi-Layer-Perception, Radial basis function, decision tree based C4.5, Bayesian networks can also provide online or offline traffic classification. These models can also classify the traffic using only the partial captures, *i.e.*, only few packets of flow and allows the reduction of data volume to be analyzed (*Jin et al., 2012*; *Luo, Xiang & Li, 2008*; *Nguyen et al., 2012*; *Singh & Agrawal, 2011*; *Zhang et al., 2019*). AdaBoost based ML models can classify the traffic based on flows rather than per packet classification. It detects the flows or transactions in the captured traffic and associates the related packets to flow as per the flow definition (*Kong et al., 2018*). Recurrent Neural Networks has the specific capability to detect the patterns from the time series data. In this regard, the long-short-term-memory based traffic model is proposed by *He, Chow & Zhang (2019)*, *Reddy & Hota (2013)* and *Zhang et al. (2019)*. These models can predict the traffic forecast on a daily basis to long term periods where the first kind of forecast is useful for the day-to-day optimizations and resource allocation and second can be used for long-term planning for the network. The convolutional neural network-based model has the capability to consider over-subscription of resources, SLA violations and can also detect and incorporate the spatio-temporal dependencies for the long-term forecast (*Bega et al., 2019*; *Reddy & Hota, 2013*). A progressive transfer learning model can provide short term forecast and long-term traffic forecasting for individual POP locations in the operator's network (*He, Chow & Zhang, 2019*). By using above ML models, traffic profiles can be dynamically learnt, and scheduling decisions are efficiently made in order to improve the QoS and network utilization. In addition to intelligent scheduling decisions, TIPS need to evaluate wireless networks with a focus on cellular networks. The traffic intensity signaling between nodes needs further research and it needs to be evaluated with other traditional algorithms also.

## CONCLUSIONS

In a detailed discussion on simulation results, we saw the TIPS approach can significantly enhance performance of a network, its utilization, and quality of services. It provides opportunities to maximize network utilization to 100% as compared to 95–97% with other approaches. We have also seen that networks exhibit better performance in regard to minimizing delay caused by queuing in CNC. Maximum enhancements to delay are several hundred percent reductions as compared to FQ, DRR, RR and SFQ. This is an excellent improvement in response time especially for real-time services and interactive services that have strict bounds for response time. We also saw TIPS provide 45% jitter reduction as compared to other approaches. We also saw that TIPS can ensure required throughput targets with the best performance and optimal network utilization. In addition to this there was no packet drop in CNC which significantly reduces wastes of network resources used to retransmission of same packets. TIPS make use of the difference in traffic intensities in different time, origin, variations to make efficient packet scheduling decisions. In CNC cases, where services performance is degraded with other existing approaches, TIPS can effectively maintain services performance, it also helps

in managing congestion by optimizing packet drop probabilities and end-to-end delay for packets which is also a factor of time-out of sessions or retransmissions. Thus, the need for minimal delay and jitter becomes very important as network diameter increases; it can dramatically provide opportunities to optimize network utilization.

### Funding
The authors received no funding for this work.

### Competing Interests
Muhammad Asif is an Academic Editor for PeerJ.

### Author Contributions
- Arif Husen conceived and designed the experiments, performed the experiments, performed the computation work, prepared figures and/or tables, authored or reviewed drafts of the paper, verification of Citations, and approved the final draft.
- Muhammad Hasanain Chaudary conceived and designed the experiments, performed the experiments, performed the computation work, prepared figures and/or tables, authored or reviewed drafts of the paper, and approved the final draft.
- Farooq Ahmad conceived and designed the experiments, analyzed the data, authored or reviewed drafts of the paper, and approved the final draft.
- Muhammad Imtiaz Alam conceived and designed the experiments, performed the experiments, performed the computation work, prepared figures and/or tables, verification of Citations, and approved the final draft.
- Abid Sohail conceived and designed the experiments, analyzed the data, authored or reviewed drafts of the paper, and approved the final draft.
- Muhammad Asif conceived and designed the experiments, analyzed the data, authored or reviewed drafts of the paper, and approved the final draft.

### Data Availability
The algorithm code, simulation code, analysis code, and automation scripts are available in the Supplemental Files.

### Supplemental Information
Supplemental information for this article can be found online at http://dx.doi.org/10.7717/peerj-cs.754#supplemental-information.

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
