# Peer review of "Improving scheduling performance in congested networks"

_PeerJ Computer Science, doi:10.7717/peerj-cs.754_

## Round 0.1 · original submission · Major Revisions

Based on reviewers’ comments, you may resubmit the revised manuscript for further consideration. Please consider the reviewers’ comments carefully and submit a list of responses to the comments along with the revised manuscript.

Reviewer 2 has suggested that you cite specific references. You are welcome to add it/them if you believe they are relevant. However, you are not required to include these citations, and if you do not include them, this will not influence my decision.

Reviewer 1 ·

Basic reporting

The authors have used two terminologies to refer to the network nodes, Serving Area Points and Point of Presence. The author needs to discuss if there are differences between the two types of nodes or they provide the same role. If the same role is provided by them, the author should use consistent terminology in different sections. Authors should note that the term POP is more common and known to the research community than SAP.

Experimental design

On page 8, line 243, authors have mentioned two types of the Erlang Distributions B and C. Authors needs to clarify what Distribution was used in the experiment and the underlying reasons.

On Page 10, Line 296 authors have used the term Traffic Intensity provided the TI values Table 3 used in experiments. Authors should clarify what mathematical models were used to calculate the traffic intensities.

Validity of the findings

On Figure 5, authors have provided the ASP traffic profiling with showing how the changes are propagated to upstream nodes. Authors needs clarify whether the symmetric or asymmetric behavior is used for the upstream and downstream traffic. Furthermore, authors need discuss the approach if the upstream and downstream data rates are different, that is the usual approach for the internet traffic.

Additional comments

It is stated on the page 10 line 315 that FOB process has been used in the experiments. Authors should clarify why this process has been used and why not the ASP or HNUB? And whether the results would be same with either of the approach.
On page 11, line 330 the terms MNC and CNC are defined, however the how theses conditions were simulated in the experiments, author needs to clarify.
On page 18, line 560 the node role has been discussed, author needs to clarify how these roles are communicated to nodes or algorithm instances.
On page 26, authors have discussed the potential use of machine learning algorithms with TIPS, authors need to clarify whether the traffic forecasting techniques proposed in the literature are capable to provide prediction of the TI values.

Reviewer 2 ·

Basic reporting

Clear and unambiguous professional English

Experimental design

Original primary research within Aims and Scope of the journal

Validity of the findings

All underlying data have been provided

Additional comments

The authors have introduced a new packet scheduling techniques based on the Erlang capacity planning methods for managing the congested networks. The results show improvement in the performance in several experiments. Following are few observations that needs to be addressed by authors.
(a) - Authors needs to clarify what advantages Erlang based TI have in comparison to the common approach based on arrival rate ,packet length and transmission rate. There are some existing packet scheduling techniques that uses the traffic intensity information to make the scheduling decisions, how this approach is different?
(b) - Two types of the Erlang B and C are mentioned. What is the effect if Erlang B or C is used? And how authors calculated during simulation?[line 243,]
(c) - the TI values given in Table 3, are these randomly chosen or what criteria authors have used to calculate them?
(d) - Authors stated that , FOB process is used, how the scheduling will work if there is no forecast information available? And how the sessions with asymmetric data rates are scheduled, authors need to provide clarification.
(e) - The result section shows presented the comparison with drr, sfq, rr and red, why these specific techniques were considered and why not others.
(f) - Authors have given the simulation topology in figure 7, Line 583. Authors need to clarify what type of traffic was used? what was the packet size and their propagation delays values? What is impact of changing these values?
(g) - Authors have shown 100% delivery ratios for the proposed scheme, what benchmarks have been considered to calculate the percentage of delivery ratios? Are these comparative or some fixed limits were used in the calculations? .
(h) - The figure 20, in the comparison section, How the overhead was calculated, was it based on actual packet captures or some other assumptions are made?
(i) - Authors need to make sure that abbreviations are consistent throughout the paper, and abbreviations must be defined where they occur first time.
(j) – It would be better if author consider to discuss following papers in section 2
Gao, Honghao, et al. "Collaborative learning-based industrial IoT API recommendation for software-defined devices: The implicit knowledge discovery perspective." IEEE Transactions on Emerging Topics in Computational Intelligence (2020).
Huang, Yuzhe, et al. "SSUR: An Approach to Optimizing Virtual Machine Allocation Strategy Based on User Requirements for Cloud Data Center." IEEE Transactions on Green Communications and Networking 5.2 (2021): 670-681.
Hussain, Walayat, and Osama Sohaib. "Analysing cloud QoS prediction approaches and its control parameters: Considering overall accuracy and freshness of a dataset." IEEE Access 7 (2019): 82649-82671.
Finally, the paper is well written there are some issues that need to address before the publication of the article.

---

## Round 0.2 · accepted · Accept

Thank you for incorporating the recommend changes. The revised version of the manuscript is now in good shape and can be published.

Reviewer 1 ·

Basic reporting

The authors have made all the requirements in the revised version. They have done all the experiments in their paper.
So, the paper looks good.

Experimental design

The authors have made all the requirements in the revised version.

All the new designs have been done in the revision.

Validity of the findings

The authors have made all the requirements in the revised version.

They have done new findings in their experiments and then added them to the revised paper.

Additional comments

The paper could be accepted for possible publication.

Reviewer 2 ·

Basic reporting

Clear and unambiguous, professional English used throughout.

Experimental design

Research question well defined, relevant & meaningful. It is stated how research fills an identified knowledge gap.

Validity of the findings

All underlying data have been provided; they are robust, statistically sound, & controlled.

Reviewer 3 ·

Basic reporting

Author have completed all requirements to make paper looks good

Experimental design

Improved design has been demonstrated in paper

Validity of the findings

This part is also improved with some new findings

Additional comments

revised version is acceptable for publication